# Individual differences in ethics positions: The EPQ-5

**Ernest H. O'Boyle**[1]*, **Donelson R. Forsyth**[2]

**1** Department of Management & Entrepreneurship, Kelley School of Business, Indiana University, Bloomington, Indiana, United States of America, **2** Jepson School of Leadership Studies, University of Richmond, Richmond, Virginia, United States of America

* oboyleeh@gmail.com

**Data Availability Statement:** Data are available at https://osf.io/jmbe5/.

**Funding:** O'Boyle's work on this project was supported by the Dale M. Coleman Chair of Management and Indiana University. Forsyth's work on this project was supported by the Leo K.

## Abstract

We revised the Ethics Position Questionnaire (EPQ), which measures variations in sensitivity to harm (idealism) and to moral standards (relativism). Study 1 identified the core components of the measured constructs theoretically and verified those features through confirmatory factor analysis (n = 2,778). Study 2 replicated these findings (n = 10,707), contrasted the theoretically defined two-factor model to alternative models, and tested for invariance of factor covariances and mean structures for men and women. Study 3 examined the relationship between the EPQ and related indicators of ethical thought (values and moral foundations) and the theory's four-fold classification typology of exceptionists, subjectivists, absolutists, and situationists. The three studies substantially reduced the original EPQ's length, clarified the conceptual interpretation of the idealism and relativism scales, affirmed the EPQ's predictive and convergent validity, and supported the four-fold classification of individuals into ethics positions. Implications for previous findings and suggestions for future research are discussed.

## Introduction

People, no matter what their personal backgrounds or cultural heritage, tend to agree when judging demonstrably benevolent or particularly egregious actions, but this consensus is lost when the discussion turns to less clear-cut issues. Do women have the right to abort their pregnancies? Should society end the life of convicted murderers? Should you press the switch to divert the trolley away from four workers, knowing that your choice will cause one person to die? Is it morally right to eat your pet dog after you accidentally run it over [1]?

Researchers have traced these variations to a number of stable individual differences, including personality traits, political ideology, cognitive development, and values [2]. The current research, drawing on ethics position theory (EPT), examines two of these sources: differences in idealism and relativism [3,4]. Morality generally involves acting in ways that minimize injuring others, but those who are less idealistic believe that harm is sometimes unavoidable. Moreover, many people rely on principles to inform their moral choices, but more relativistic individuals are skeptical about the possibility of formulating universal moral standards. The Ethics Position Questionnaire (EPQ) measures these two dimensions of individuals' moral philosophies.

and Gaylee Thorsness Chair in Ethical Leadership
and University of Richmond.

**Competing interests:** The authors have declared
that no competing interests exist.

The EPQ has proven useful in predicting differences in moral judgments in a variety of contexts [5–7]. The empirical record, however, is not pristine. Some researchers have not been able to confirm the EPQ's two-factor structure. Others find some scale items have low factor loadings or weak correlations with total scores—often resulting in their deletion to achieve satisfactory psychometrics. Further, associations between the two scales and related constructs, such as moral values, do not consistently support the theory's predictions, including the presumed conjoint interaction of the two indexes.

We addressed these concerns by conducting an extensive analysis and revision of this instrument. We first briefly review previous studies of the EPQ's psychometric properties, and based on that review, propose a new (and shorter) version of the inventory: the EPQ-5 (S1 Table). We then examine the revised instrument's adequacy in two large samples and conduct a measurement invariance analysis to ensure that the EPQ's latent factor structure is valid for male and female respondents. We also examine the rationale for and evidence pertaining to the classification of individuals into one of the four positions identified by the theory, and report evidence of the convergent and divergent validity of the EPQ-5.

## Ethics position theory

Both philosophical and psychological analyses of morality have consistently identified two critical determinants of moral decision-making: the degree of harm or benefit produced by the action and the consistency of the action with standards that define what is considered moral. Philosophers have traditionally contrasted moral theories based on principles (deontological models) and models that stress the consequences of actions (teleological models). Similarly, psychological researchers (e.g., [1,8,9]) have explored the interplay between rule-based judgment and sensitivity to welfare of others.

Ethics position theory agrees that actions that cause harm or are inconsistent with commonly accepted standards of morality trigger moral scrutiny, but the theory maintains that people differ in their orientation toward both influences. Some people are more idealistic than others, for they believe that "If an action could harm an innocent other, then it should not be done." Some people, too, believe that "what is ethical varies from one situation and society to another," but others disagree with such moral relativism. Schlenker and Forsyth [10] initially identified these two themes in their study of people's moral judgments of Milgram's [11] studies of obedience, and Forsyth [3] extended these findings by developing the EPQ to measure both idealism and relativism. The idealism items stressed humanitarian, visionary ideals, such as avoiding others' harm, whereas the relativism scale includes items that question the value of relying on moral rules or codes of ethics when making moral judgments.

**Ethics positions.** Ethics position theory assumes people range along two continua defined by their degree of idealism and relativism, but it also suggests individuals' positions along each continuum defines their ethical type. The two-dimensional model of ethical positions shown in Table 1 considers both principles (relativism) and outcomes (idealism), only principles,

**Table 1. The four moral types identified in EPT.**

|  | **Low Relativism** | **High Relativism** |
|---|---|---|
| Low idealism | **Exceptionists**: Conventionalists who tolerate exceptions to moral standards when benefits offset potential harmful consequences. | **Subjectivists**: Realists who do not endorse moral standards that define right and wrong or the avoidance of harmful consequences |
| High idealism | **Absolutists**: Principled idealists who endorse both reliance on moral standards and striving to minimize harm done to others | **Situationists**: Idealistic contextualists who value minimizing harm rather than reliance on moral standards that define right and wrong |

only outcomes, or neither. *Exceptionists* are relatively conventional in their moral orientation. They disagree that morality is purely personal but recognize that moral principles do not always minimize harm. *Subjectivists* are skeptical about basing moral judgments on trans-situational and trans-personal rules (high relativism), but they do not strongly endorse the "do no harm" mandate (low idealism). *Absolutists*, unlike both the exceptionists and the subjectivists stress harm minimization, but as the name suggests, endorse moral standards. *Situationists*, in contrast, do not believe that moral standards provide a bright line between what is morally good and bad (relativism), but they are humanitarian in their orientation—they are committed to promoting human well-being.

**Factor structure of the EPQ.** The EPQ items were selected to measure idealism and relativism—and only idealism and relativism. A number of investigators have tested this assumption using exploratory or confirmatory factor analysis (e.g., [12–26]). These investigations generally confirmed the EPQ's two-dimensional structure, but some studies suggested the original EPQ could be improved through additional scaling and psychometric refinement. Several studies, too, suggested that certain items on the EPQ are not linked to either idealism or relativism, but instead formed a coherent subcluster within these two domains. For example, Davis and colleagues [17], using confirmatory factor analysis, identified a third factor which included the two items that pertained specifically to lying (veracity). Moreover, several investigators, after administering the EPQ, found that certain items were not highly correlated with the other items, and so removed those items to achieve acceptable internal reliability. When Forsyth et al. [4] meta-analytically reviewed a set of 83 studies that used the EPQ between 1980 and 2006 with 140 different samples, they identified 25 (18.1%) that used a shortened version of the idealism scale, and 30 (22.9%) that used a shortened version of the relativism scale. In consequence, observed differences in results across studies may be due to true differences in participant features or study contexts (e.g., occupation, nationality, generational cohort), but these differences may simply be statistical artifacts that result from the use of altered versions of the EPQ.

## Overview of studies

Prior research leaves unanswered several significant questions about the inventory that measures these variations—the EPQ—and the taxonomy of moral types it describes. The EPQ was designed to assess two latent constructs—idealism and relativism—but previous researchers have not consistently confirmed the measure's psychometric and conceptual adequacy.

We sought to reduce this uncertainty in the current research by conducting an empirically driven revision and validation of the EPQ. In Study 1, we evaluated the EPQ's factor structure, item adequacy, scale length, and psychometric properties to determine the most accurate and efficient means by which to measure individual's ethics positions. In Study 2, we replicated the factor structure, tested the EPQ for measurement invariance across men and women, and used the revised measure to classify individuals into the four proposed categories. In Study 3, we gathered information about the convergent and divergent validity of the EPQ. The EPQ has been used by several researchers in their studies morality and moral judgment, and the findings are generally support the validity of both the idealism and relativism scale. However, the relationship between the EPQ and other measures of individual differences in moral values requires further investigation. If the EPQ is a valid measure of people's level of idealism and relativism, and these two sets of beliefs influence how they think about moral issues, then the EPQ should successfully predict individual differences in related moral beliefs and values, including Schwartz's [27] social values scale and the moral foundations questionnaire developed by Haidt and his colleagues (e.g., [28,29]).

## Study 1: Scale revision

Previous studies of the factor structure of the EPQ indicate the inventory adequately measures individual differences in idealism and moral relativism, but that the original questionnaire could be improved to increase its reliability and validity. We therefore revised the EPQ, seeking two related, but not entirely compatible, goals. First, to ensure construct fidelity, we reviewed each item's association with its respective construct—avoiding harming others (idealism) and skepticism about the usefulness of inviolate, trans-situational rules when making moral judgments (relativism)—to make certain the revised scale sampled fully the identified construct domains. Second, to ensure construct validity and maximize the predictive power of the measure, we eliminated any items that introduced novel elements that were not directly related to those constructs.

### Methods and materials

**Participants.** We used a subset of respondents who completed an English-language version of the original EPQ at an online survey site and the data are publicly available at https://osf.io/jmbe5/. These individuals accessed the site primarily through web searches, and their only incentive was learning their scores on the idealism and relativism scales. The respondents were residents of the US (58.8%), Netherlands (8.0%), UK (7.7%), Sweden (5.0%), Australia (4.4%), Philippines (2.9%), Spain (2.4%), Canada (2.1%), and over 50 other nations (10.8%).

We used best practice recommendations to screen data for insufficient effort/careless responding [30]. Specifically, we first screened out those participants who did not complete at least 80 percent of the EPQ and had duplicate IP addresses. We also conducted a max run analysis, which flags any participant that gives the same response for an extended series of items. Using the frequency distribution as a guide, we determined a significant gap in consecutive responding at seven (i.e., very few individuals gave the same level of agreement for seven or more consecutive items). Finally, we screened for multivariate outliers using Mahalanobis distances with a conservative $p$-value cutoff of .001. The final sample size was 2,778. Because listwise and pairwise deletion result in biased estimates and less accurate hypothesis tests [31], for those with incomplete data (i.e., missing between one and four items of the EPQ), we used the R package MICE [32] with predictive mean matching to impute missing scores.

**Confirmatory factor analyses (CFA).** We examined the EPQ's theoretically specified factor structure and alternative specifications in a series of CFAs in the R package lavaan [33]. Overall model fit was assessed based on the three recommended global fit indices; comparative fix index (CFI; [34]), root mean square error of approximation (RMSEA; [35]) with its 90% confidence interval, and the standardized root mean square residual (SRMR). Consistent with recommendations for model evaluation [36], we supplemented the global fit tests with an examination of composite reliabilities and the strength of the standardized factor loadings.

### Results and discussion

The CFA results are presented in Table 2 and more detailed discussion of the various model specifications are provided in the Electronic Supplemental Materials. The full, 20-item scale results of the EPQ were middling whether modelled as the theorized two-factor solution (Two-factor-I & R) or with the inclusion of veracity modelled as an additional factor (Three-factor-I, R, & V). The RMSEA and SRMR were generally adequate, but CFI values all fell under the .95 threshold [37]. Average variance extracted (AVE) for these models was also marginal and a number of factor loadings in both the two and three factor full scale specifications were less than .40. Modification indices noted sizable, correlated residuals for several of the items.

**Table 2. Confirmatory factor analyses for Study 1.**

| | $\chi^2$ | df | CFI | RMSEA | 90% CI | SRMR |
|---|---|---|---|---|---|---|
| Full scale | | | | | | |
| One-factor | 5991 | 170 | .542 | .111 | .109; .113 | .113 |
| Two-factor-I & R | 2934 | 169 | .782 | .077 | .074; .079 | .070 |
| Three-factor-I, R, & V | 1534 | 167 | .892 | .054 | .052; .057 | .049 |
| Two-factor-I & R, no V | 1377 | 134 | .884 | .058 | .055; .061 | .051 |
| EPQ-5 | 344 | 34 | .951 | .057 | .052; .063 | .041 |

Note: I = Idealism, R = Relativism, V = Veracity, df = degrees of freedom, CFI = comparative fit index, RMSEA = root mean square of approximation, CI = confidence interval, SRMR = standardized root mean square residual.

Excluding the two items in the relativism scale that pertained to lying/veracity (Two-factor-I & R, no V) did little to improve overall model fit. In particular, the number and magnitude of the correlated residuals among several items indicated that some of the items not only share variance that is attributable to latent levels of idealism and relativism, they also share variance attributable to external artifacts. These external artifacts can be substantive (e.g., two items prone to socially desirable responding) or a function of the measurement (e.g., two overly complicated items that might elicit uncertain responses at the midpoint of the Likert scale).

**Item analysis.** As the CFA findings would suggest, item analysis indicated the two scales of the EPQ lacked sufficient internal consistency: despite their content being aligned with EPT, the signal to noise ratio for the 18 items that remained after eliminating the veracity items was unacceptable, and merited further EPQ refinement. We therefore examined each item's content to ensure its alignment with EPT as well as consulting past research on the EPQ's dimensionality. We investigated the five items on the idealism scale and three items on the abridged relativism scale with the lowest factor loadings. Upon examining these items, they included content that was narrow in scope or not clearly related to the conceptual meaning of the scale. For example, the primary items on the idealism scale emphasize harm avoidance (Items 1, 3, 4, and 6). Item 2, however, raises the issue of risk—"risks to another should never be tolerated"—but the word *risk* can mean very different things to different people, including uncertainty, hazard, probability and even danger [38]. Similarly, Item 10 suggests one strive to reach the ideal of "perfect action," and makes no mention of harm at all. Several other items on the relativism scale were similarly narrow in scope or include extraneous content, such as Item 18: "Rigidly codifying an ethical position that prevents certain types of actions could stand in the way of better human relations and adjustment." This item, in addition to using loaded wording ("rigidly codifying") does not pertain to the core focus of the relativism inventory: skepticism regarding moral rules. Other low-loading items were wordy and cognitively demanding (e.g., "Deciding whether or not to perform an act by balancing the positive consequences of the act against the negative consequences of the act is immoral"). We speculate that these features were the reasons for their poor loadings, correlated residuals, and were likely the primary contributing factor to the middling overall model fit.

Both idealism and relativism are unidimensional constructs, thus these items do not represent meaningful facets of idealism or relativism. Rather, they are imperfect indicators of overarching constructs. Based on the collective evidence, we concluded that their exclusion would not result in a loss of construct validity, create a deficiency bias, or detriment the EPQ's other psychometric properties. The result of this scale revision was that idealism and relativism would both be measured with five items each (henceforth referred to as the EPQ-5, to indicate the number of items on each subscale and each subscale is a distinct measurement index).

**Table 3. Psychometrics for EPQ for Study 1 (left) and Study 2 (right).**

|  | factor loading | item-scale r | alpha if deleted | CR | AVE | α |
|---|---|---|---|---|---|---|
| Idealism |  |  |  | .80/85 | .45/.53 | .80/.85 |
| Q1 | .62/.75 | .55/.68 | .78/.81 |  |  |  |
| Q3 | .63/70 | .57/.64 | .77/.82 |  |  |  |
| Q4 | .75/.74 | .63/.66 | .75/.82 |  |  |  |
| Q5 | .76/.77 | .64/.69 | .75/.81 |  |  |  |
| Q6 | .60/.69 | .54/.63 | .78/.83 |  |  |  |
| Relativism |  |  |  | .70/.81 | .33/.47 | .70/.81 |
| Q12 | .46/.61 | .37/.54 | .68/.79 |  |  |  |
| Q13 | .72/.80 | .57/.70 | .59/.74 |  |  |  |
| Q15 | .70/.79 | .36/.69 | .68/72 |  |  |  |
| Q16 | .50/.63 | .55/.55 | .60/.79 |  |  |  |
| Q17 | .42/.56 | .43/.40 | .65/.80 |  |  |  |

Note: Q = Corresponds to the retained items from the original EPQ, CR = composite reliability, AVE = average variance extracted.

**The EPQ-5.** A CFA of the shortened EPQ (Table 2) indicated the EPQ-5 performed exceptionally well and exceeded all traditional thresholds (CFI = .951, RMSEA = .057 (.052; .063), SRMR = .041). Table 3 provides more detailed information at the item and factor level. All factor loadings exceeded .40 and all but one item was .50 or greater. The deletion of any item resulted in a decrease in Cronbach's alpha, and despite composite reliability being strongly influenced by the number of items, composite reliability decreases were minimal for the EPQ-5 relative to the full version ($\Delta$ idealism = .01, $\Delta$ relativism = .03). Further, compared to the original scale, the average variance extracted (AVE) improved greatly for both idealism and relativism (AVE-Idealism = .45; AVE-Relativism = .33).

The instructions and items for the EPQ-5 are included in the Appendix. The mean and median scores for this sample were 3.83 ($sd$ = 1.06) and 4.0 for idealism, and 3.22 ($sd$ = 1.15) and 3.0 for relativism, and the scores show a slight negative skew; -0.77 and -0.17, respectively. Both scales are internally consistent ($\alpha$s = 0.80 & 0.70, respectively) and both correlate with the longer, original measures of idealism and relativism; $r$s = 0.93 and 0.89, respectively. As with the original scale and consistent with EPT, the correlation between idealism and relativism scores was only -0.04.

## Study 2: Replication and classification

The revised EPQ developed in Study 1 focused more clearly on the latent constructs of idealism and relativism, internal consistency of the two scales, and the proposed two-factor (idealism and relativism) solution. In Study 2 we (a) replicated our psychometric analysis with a larger sample, (b) examined the EPQ-5's measurement invariance across men and women, and (c) how well the EPQ-5 classified individuals into the four ethical types identified in EPT.

### Methods and materials

**Participants.** We secured a sample of 11,917 individuals from the online assessment site YourMorals.org. The respondents were volunteers who usually found YourMorals.org "through publicity about psychological research or by typing keywords related to morality into an Internet search engine" [39, p. 3]. The respondents were residents of the US (73.0%), Canada (5.4%), UK (5.2%), Australia (2.1%), Germany (0.9%), France (0.5%) and over 100 other nations (18.1%). The same data cleaning procedures employed in Study 1 were repeated here

with the additional screening of those that did not indicate their sex, which was needed for the tests of measurement and structural invariance. The final sample size was 10,252.

## Results and discussion

**Factor and item analysis.** Confirmatory factor analysis confirmed the adequacy of the revised version of the EPQ, for a two-factor model fit was excellent: $\chi^2(21) = 678.7$, CFI = .983, RMSEA = .042 (.039; .045), SRMR = .038. Of note, the upper bound of the 90% RMSEA confidence interval is below .05, thus we support the close-fit hypothesis [40] and retain the null hypothesis of perfect fit.

The mean and median scores for this sample differed very little from that of Study 1: (Idealism: M = 3.49, Mdn = 3.6, *sd* = .92, Relativism: M = 3.10, Mdn = 3.1, *sd* = .96). The distribution of the scores is slightly skewed; -0.45 and -0.13, respectively. Both scales' Cronbach's alphas exceed 0.80 and the composite reliability and average variance extracted are acceptable. As before, the correlation between idealism and relativism scores for the EPQ-5 was trivial ($r = 0.05$).

**Measurement invariance across the sexes.** When we compared the responses of the 4,087 women and 6,153 men who completed the EPQ-5, we discovered the women's average scores for both idealism and relativism were higher than the men's scores; $Fs(1, 10,235) = 524.34$ and 23.57, $ps < .001$. The idealism and relativism means (and medians) were 3.73 (3.70) and 3.15 (3.10) for women and 3.32 (3.30) and 3.06 (3.00) for men. However, it is unclear whether these mean differences reflect true differences between the sexes on idealism and relativism or differences in interpretation of the EPQ-5. Determining this requires tests of measurement and structural invariance (MI/SI).

Using procedures outlined elsewhere [41,42], we tested for configural, metric, scalar, and residual invariance tests for MI and factor variance and factor mean invariance tests for SI were conducted in Mplus 8.0 [43]. The summary fit indices are presented in Table 4, the standardized factor loadings for men and women are presented in Fig 1, and the full details and model results are reported in the Electronic Supplemental Materials.

The EPQ-5 demonstrated *strict measurement invariance* meaning that men and women had equivalent factor loadings, intercepts, and error variances. That is, observed differences in item means between men and women were attributable to factor (true score) mean differences only. Regarding structural invariance, men had slightly larger variances in both relativism and idealism, and women had significantly higher latent mean levels of idealism (women higher).

**Table 4. Tests of measurement invariance and structural invariance (Study 2).**

| Constraint/Test | $\chi^2$(df) | CFI | RMSEA | 90% CI | SRMR |
|---|---|---|---|---|---|
| Configural | 691(68) | .982 | .042 | .039; .045 | .038 |
| Women | 270(34) | .983 | .041 | .037; .046 | .035 |
| Men | 421(34) | .982 | .043 | .039; .047 | .035 |
| Metric | 789 (78) | .980 | .042 | .040; .045 | .053 |
| Scalar | 889(84) | .977 | .043 | .041; .046 | .039 |
| Residual Variance | 1219(94) | .968 | .048 | .046; .051 | .043 |
| Factor Variance | 1301(96) | .965 | .050 | .047; .052 | .061 |
| **Relative Mean Only** | **1329(97)** | **.964** | **.050** | **.047; .052** | **.062** |
| Both Factor Means | 1832(98) | .950 | .059 | .056; .061 | .083 |

Note: Boldface indicates retained model. df = degrees of freedom, CFI = comparative fit index, RMSEA = root mean square of approximation, CI = confidence interval, SRMR = standardized root mean square residual.

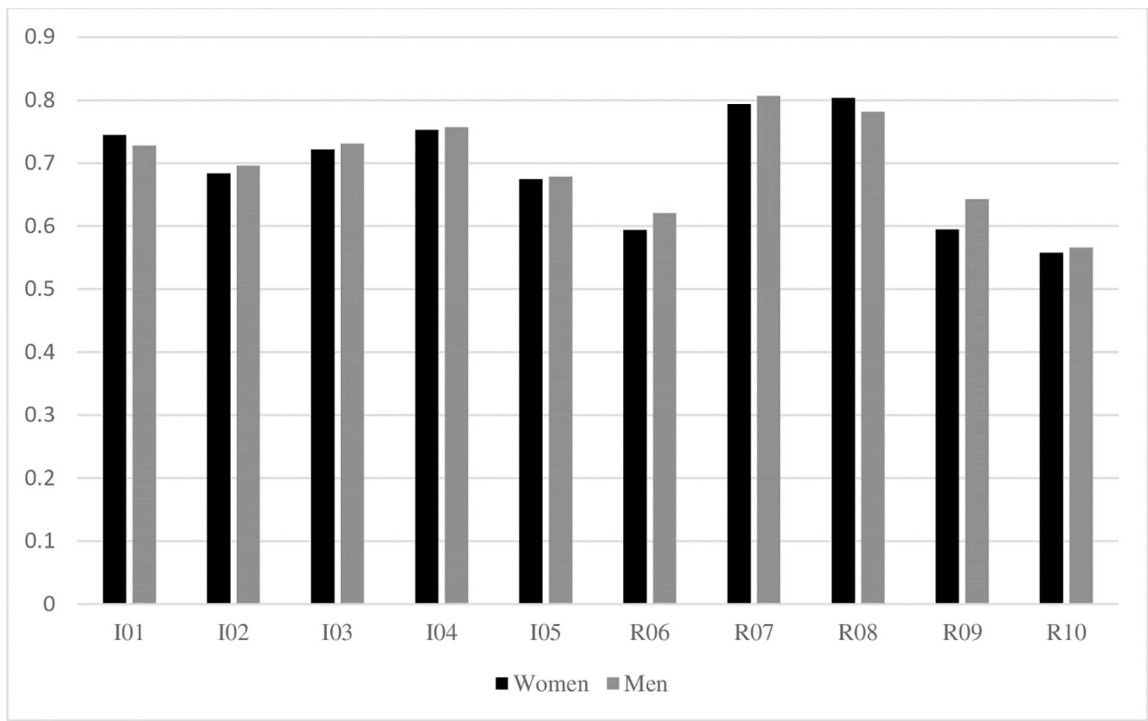

**Fig 1. Standardized factor loadings across sexes.** Note: I = Idealism, R = Relativism. Item numbers correspond to the ten items of the EPQ-5 supplied in the Appendix.

Latent mean differences for relativism were negligible. Using decision rules outlined in Chen [44], the retained model was the fully constrained measurement model that only allowed latent mean differences in idealism to vary ($\chi2(97) = 1329$, CFI = .964, RMSEA = .050 [.047; .052], SRMR = .062). We conclude that the EPQ-5 is equally interpreted across the sexes, women are higher in idealism than men, and regarding MI and SI, the EPQ-5 is psychometrically valid.

**Classification analysis.** Ethics position theory assumes differences in moral thought are determined, in part, by differences in idealism and relativism, but it also suggests that these two dimensions combine to define a person's ethics position. As Table 1 summarized, situationists are relativistic and idealistic, subjectivists are relativistic but not idealistic, absolutists are not relativistic but idealistic, and exceptionists are neither relativistic nor idealistic. These moral types are assumed to be qualitatively different from one another and, to some extent, mutually exclusive.

Although the theory predicts that individuals' level of idealism and relativism determines their distinct ethics positions, the four groups identified in Table 1 may not correspond to the natural groups that exist within the population. Rather than four categories, people may naturally cluster into fewer categories, more categories, or no categories at all. Fortunately, researchers have developed several statistical procedures to identify clusters of individuals within a more general population, including Bayesian expectation maximization algorithms, cluster analysis, latent class modeling, and taxonometric procedures [45]. We used one of these methods, cluster analysis, to compare the theory's recommended taxonomy of ethics positions to the naturally occurring distribution of individuals into cluster or subgroups. A high degree of correspondence between these two classification methods would provide support for the theory's type-based approach to ethics positions.

**Table 5. Ethics position as determined by relative idealism and relativism scores (EPQ classification) and by cluster analysis.**

| Ethics Position | Cluster Analysis Classification | | | | Total | % Accuracy |
|---|---|---|---|---|---|---|
| | **Exceptionist** | **Subjectivist** | **Absolutist** | **Situationist** | | |
| Exceptionist | **2012** | 0 | 285 | 47 | 2344 | 85.84 |
| Subjectivist | 0 | **1666** | 0 | 322 | 1988 | 83.80 |
| Absolutist | 0 | 0 | **1903** | 302 | 2205 | 86.30 |
| Situationist | 0 | 0 | 0 | **2155** | 2155 | 100.00 |
| Total | 2012 | 1666 | 2188 | 2826 | 8692 | |

*Classifications into types*. We first classified each individual, a priori, into one of the four ethics positions using a normative procedure. Individuals whose idealism and relativism scores were above average, with the average defined by the median for the sample (3.6 and 3.1), were considered to be high idealists and high relativists. Those with scores below the median were classified as low idealists and relativists. The number of respondents in each ethics position were nearly equal: 27% were classified as exceptionists, 25% as absolutists or situationists, and 23% as subjectivists (see Table 5).

We used hierarchical cluster analysis (HCA, Ward's technique) to identify naturally existing subgroups within this sample. This method's agglomerative algorithm calculates similarities, based on Euclidian distances, between possible clusters. It initially classifies the most similar persons into clusters, and then at each subsequent step identifies those individuals who are most siilar to those clusters. The final step in the algorithm combines all clusters.

Inspection of the magnitude of the squared Euclidean distances between the clusters combined in the final 12 steps in this process suggested that a 4-cluster solution best fit the data. The magnitude of the distances remains relatively constant as clusters are combined into fewer and fewer clusters, until 4 clusters are combined into three. This step generated the largest acceleration of distance growth—the "elbow" indicating where to stop combining clusters (see the Electronic Supplemental Materials).

*Degree of Congruence*. The 4-group classification generated through cluster analysis was congruent with the a priori classification of individuals based on their ethics positions. The two classifications matched for 89.0% of the cases, although accuracy varied across the four ethics positions. The two methods of classification agreed for 85.8% of the exceptionists, 83.8% for subjectivists, 86.3% for absolutists, and 100% for the situationists. Discrepancies, when they did occur, usually resulted from differences on one rather than two of the EPQ factors. For example, 302 individuals were classified as absolutists by one method and situationists by the other. Both of these ethics positions are idealistic ones, although they differ in relativism. Similarly, the two classification methods were generally congruent in their classification of the respondents who were relativists (situationists and absolutists), but they diverged because one identified 322 as subjectivists but the other classified them as situationists.

## Study 3: Moral thought

The results of Studies 1 and 2 confirm the psychometric adequacy of the two scales of the EPQ, and the scale's content validity is more certain given it includes only items that directly pertain to the core concepts the EPQ was designed to measure: idealism and relativism. Psychometric adequacy, however, does not insure theoretical adequacy. Although a number of studies of the original EPQ suggested that its two scales were appropriately associated with measures of similar psychological constructs, we conducted a third study to determine the validity of EPQ-5 measures, using two related measures of moral values and beliefs as criteria: the Schwartz Values Scale (SVS; [46–48]) and the Moral Foundations Questionnaire [28,29,49]).

Schwartz's Value Survey assesses ten values that are common in most cultures: power, achievement, hedonism, stimulation, self-direction, universalism, benevolence, tradition, conformity, and security. He arranges the values in a circumplex model based on compatibility and incompatibility of these values, with values varying along two dimensions: self-transcendence (e.g., universalism, benevolence) vs. self-enhancement (e.g., power, achievement) and openness to change (self-direction, stimulation) vs. conservation (e.g., tradition, conformity). Some of the values Schwartz identifies and assesses are theoretically congruent with variations in idealism or relativism [50,51]. Specifically, idealism's emphasis on minimizing harm to others is consistent with the self-transcendence values of benevolence and universalism. Idealists are harm-averse, and so should be more likely to agree that "working for the welfare of others" and is an important value, as is a "willingness to pardon others" for their offenses (benevolence). On the other hand, relativists should be less likely to endorse values that are more conservative. Relativists are skeptical about the importance of relying on standards when making moral judgments and their rejection of traditional perspectives suggests they would be disinclined to agree with such values as "courtesy," "good manners," and "self-restraint" (conformity) or "preservation of time honored customs" and "holding to religious faith" (tradition).

The Moral Foundation Questionnaire, developed by Haidt and his colleagues, assesses five universal moral foundations: harm, fairness, loyalty, authority, and sanctity (28, 29, 49). Although it should be noted that Haidt and his colleagues continue to refine the labels and definitions of each of the five foundations. The original labels were harm/care, fairness/reciprocity, ingroup/loyalty, authority/respect, and sanctity/purity. A sixth foundation, liberty/oppression, is also being considered for inclusion in the theory [29]. Although both the EPQ and MFQ trace individual differences in morality back to differences in tolerance of actions that cause harm to others and differences in respect for traditional, culturally approved conceptions of what is moral and what is not, the theories differ in the number of components that they use to fully explain these variations in moral judgment. Ethics position theory favors two—idealism and relativism—whereas MFT includes (at least) five: harm, fairness, loyalty, authority, and sanctity. However, the harm and fairness foundations in MFQ are often closely associated with one another, as are the ingroup, authority, and sanctity foundations [52]. In MFQ, harm and fairness are the individualizing foundations, since they serve to protect the personal rights and well-being of individuals in society. In contrast, loyalty, authority, and sanctity are the binding foundations, for they "are about binding people together into larger groups and institutions" ([52], p. 369). These convergences suggest that idealism will be more closely associated with the individualizing foundations, whereas relativism will be associated with the binding factors.

## Methods and materials

**Participants and measures.** The participants in this phase of the research included all individuals who took part in the YourMorals project and completed both the EPQ and one or more measure of moral attitudes, values, or traits. Because respondents selected which questionnaires they wished to complete from the site, the sample sizes vary. The same data cleaning procedures employed in Study 1 were repeated here and sample size ranges are reported in Table.

*Schwartz Values Scale (SVS).* The Schwartz Value Survey [27,53] identifies ten values as fundamental and possibly universal. Respondents are asked to rate each item from -1 to +7, where 0 indicates this value is of low importance for the person and a 7 of supreme importance. A rating of -1 can be given to indicate opposition to the value. We calculated scores for the four primary dimensions of his theory—self-transcendence, conservation, openness, and self-

enhancement—and the one specific value that is not included in the four primary value dimensions: hedonism.

*Moral Foundations Questionnaire (MFQ).* The MFQ assesses the five bases of morality described by moral foundations theory: harm, fairness, loyalty, authority, and sanctity. Respondents indicate their degree of agreement with statements that are consistent with each foundation, including "Compassion for those who are suffering is the most crucial virtue" (harm), "Just is the most important requirement for a society" (fairness), "People should be loyal to their family members, even when they have done something wrong" (loyalty), "Respect for authority is something all children need to learn" (authority), and "I would call some acts wrong on the grounds that they are unnatural" (sanctity; [52], p. 385).

## Results and discussion

The correlations among idealism and relativism—as measured by the original EPQ and the shortened EPQ-5—and dimensions of moral values assessed by the SVS and the moral foundations assessed by the MFQ are presented in Table 6 (controlling for gender). These correlations provide support for the adequacy of the EPQ-5, as well as evidence of convergent and divergent validities.

*Concurrent validity.* The correlations between the scores on the SVS and MFQ and the EPQ's idealism and relativism scores were consistently larger for the original EPQ in comparison to the EPQ-5, but as Table 6 indicates, the differences were relatively unsubstantial, ranging from .01 to .05. These correlations suggest that the revised EPQ, despite including fewer items, yields results that are essentially equivalent to the original inventory. The correlations between the original and revised idealism and relativism scales were .94 and .93, respectively.

*Convergent and divergent validity.* The correlations in Table 6 indicate that idealism was more closely associated with the self-transcendence values (universalism & benevolence) of the SVS than conservation or self-enhancement and was not associated with either openness or hedonism. Relativism, in contrast, was negatively correlated with conservation, and more substantially correlated with both openness and hedonism. Table 6 also indicates that idealism was more closely associated with the harm and fairness scales of the MFQ than with the three factors of loyalty, authority, and sanctity (all $ps < .001$ given the large sample sizes). Relativism, in contrast, was more substantially correlated with loyalty, authority, and sanctity than with harm and fairness.

**Table 6. Correlational comparison of the original EPQ and the revised EPQ (EPQ-5) in predicting associations with moral values (SVS: Schwartz Value Scale) and moral foundations (MFQ: Moral Foundations Questionnaire).**

| Variable | Idealism | | Relativism | |
|---|---|---|---|---|
| | EPQ-5 | EPQ | EPQ-5 | EPQ |
| SVS: Self-Transcendence | .42 | .43 | .03 | .01 |
| SVS: Conservation | .17 | .14 | -.20 | -.24 |
| SVS: Openness | .07 | .05 | .17 | .18 |
| SVS: Self-Enhancement | -.14 | -.14 | -.02 | -.01 |
| SVS: Hedonism | .00 | -.01 | .23 | .25 |
| MFQ: Harm | .53 | .54 | .02 | .03 |
| MFQ: Fairness | .38 | .39 | .06 | .07 |
| MFQ: Loyalty | -.08 | -.06 | -.14 | -.17 |
| MFQ: Authority | -.11 | -.08 | -.21 | -.24 |
| MFQ: Sanctity | .03 | .08 | -0.33 | -.38 |

Note: N = 3,765 for correlations between EPQ and SVS and N = 8,635 for correlations between EPQ and MFQ.

*Ethics positions*. We also examined the relationship between the two scales of the EPQ and individuals' values in two separate 2 (idealism: low and high) X 2 (relativism: low and high) X 2 (sex: men and women) MANOVAs—one for the SVS variables and one for the MFQ variables—with age as a covariate. These results, which are presented in the Electronic Supplemental Materials, are consistent with the correlational analyses presented in Table 6. Individuals who adopted ethics positions that are idealistic—absolutists and situationists—differed from more pragmatic individuals—exceptionists and subjectivists—for all the SVS moral values (except hedonism) and all the MFQ foundations (except purity). Individuals who adopted relativistic ethics positions—the situationists and subjectivists—differed from the nonrelativists—absolutists and exceptionists) in the conservation, openness, and hedonism moral values, and the MFQ dimensions of loyalty, authority, and purity. However, the interaction of sex, idealism, and relativism was significant at the multivariate level for the SVS moral value measures, in some cases qualifying the more general effects of relativism and idealism. For example, both idealistic men and women had higher self-transcendence values than low idealists, but exceptionist men were particularly low in self-transcendence, whereas situationist women scores were substantially elevated for this value. In contrast, relativists were more open than nonrelativists, but situationist women had particularly elevated scores on openness, particularly in comparison to exceptionist men. In contrast, the interaction of idealism and relativism was not qualified by sex in the analysis of the MFQ items. As the correlational analyses suggested, the main effect of idealism was more substantial for harm and fairness, whereas the main effect of relativism was more robust for loyalty, authority, and sanctity. In addition, the interaction of idealism and relativism was significant only for loyalty, authority, and sanctity. Exceptionists were not harm or fairness oriented, but more likely than all other ethics positions to value loyalty, authority, and sanctity. Subjectivists matched exceptionists' relative disinterest in harm and fairness, but they also did not value loyalty, authority, and sanctity; they valued sanctity lower than all other groups. Absolutists and situationists concurred in their emphasis on harm and fairness but diverged in their weighing of loyalty and authority; absolutists were average, overall, in their emphasis on loyalty and authority, whereas situationists were below average. The two sets of idealists, however, diverged more substantially on sanctity; absolutists endorsed sanctity as a moral foundation, whereas situationists did not.

## General discussion

The Ethics Position Questionnaire has been consistently applied across a variety of psychological disciplines and related fields (e.g., marketing, economics, organizational behavior) but despite this widespread use, questions remained about its psychometric adequacy, construct validity, and the appropriateness of the taxonomy. The current investigation examined these questions by refining an individual difference measure of these tendencies and confirming the validity of the proposed measure by using it to predict differences on related measures of moral values and foundations. The results of Studies 1 and 2 provided reassuring evidence of the psychometric adequacy of the two scales of the EPQ-5. Study 1 confirmed the proposed measurement model. Study 2 provided further support in a larger data set by demonstrating that the shorter measure of idealism and relativism provided excellent fit to the 2-factor measurement model. Further, we provided, to our knowledge, the first test of the EPQ's measurement invariance across men and women and concluded that the higher levels of idealism reported by women is not due to differences in item interpretation, but to women's higher levels of concern for harm compared to men.

Study 3's results support the EPQ's construct validity, for they indicate the two subscales of the EPQ—idealism and relativism—actually measure individual differences in sensitivity to

the level of harm that that an action may cause and the consistence of the action with moral absolutes that serve as guides to action and judgment, and are related—as theory suggests they should be—to individual differences in values [54] and moral foundations [52].

Study 3's findings also provide suggestive, but not altogether compelling, support for EPT's claims regarding moral types based on variations in idealism and relativism. The types are generated by determining a person's position on two or more dimensions, which are crossed in a cross-classification scheme. Although dichotomizing continuous variables is generally considered "a bad idea" [55], "a practice to avoid," [56], and therefore "inadvisable" [57], a typology approach is warranted in this case on both theoretical and empirical grounds. Philosophical analyses of ethical thought have, by tradition, identified discrete positions regarding moral judgment, such as deontology and utilitarianism. Ethics position theory, in extending this assumption to the analysis of psychological perspectives, maintains that individuals' sensitivity to harm and principles combined to generate a coherent ethical ideology. Empirically, Study 2 used classification analysis to confirm the congruence between this classification method and the clustering of respondents into naturally occurring categories. In Study 3, the ethical types differed, systematically, as well.

## Study implications

Just as reliability is the ceiling for validity, the ceiling for theoretical advances is the accuracy of measurement. The methodological, empirical, and theoretical implications of the research reported here are considerable. Focusing on methodology, this study both affirms the psychometric adequacy of the EPQ, but also refines the measure to index individual differences more accurately in moral thought. The revised scale, given its length and the removal of items that were not reliably associated with the scales' underlying constructs, will offer researchers the means to assess ethics positions more easily and also reduce the variation in the instrument's content across studies. As stated above, a common practice with the full length EPQ is the dropping and altering of items [4]. This is often engaged in to reduce survey lengths and/or improve the reliability scale. However, this distorts a measure's psychometrics as well as impedes the robustness of systematic reviews that cannot tease apart variance attributable true score differences from variance attributable to differences in the measure [58]. The EPQ-5 provides an already concise measure and one that does not and should not require further reduction.

Empirically, the findings support the central tenets of EPT. We found that idealism and relativism are coherent psychological constructs, and the EPQ-5 accurately measures these constructs. Moreover, the findings also support the theory's prediction of both a direct and a configural effect of one's level of idealism and relativism on moral thought, including moral values and foundations. Variations in idealism and relativism are independently associated with values and foundations, but in some cases their relationship to this outcome is a conjoint one. These interactive effects of idealism and relativism provide indirect justification for the taxonometic assumptions of the theory.

Theoretically, the findings are congruent with recent advances in moral psychology and provide the means to more fully integrate those efforts. Despite the critical importance of morality for maintaining stable interpersonal relationships in human societies, disagreement over what is moral and what is immoral is common. Many factors contribute to this diversity, but the current work identifies one significant factor: differences in ethics positions. This analysis is consistent with recent advances in moral psychology, particularly the focus on individual differences that have been examined extensively in philosophical analyses of ethics. Ethics position theory explicitly identifies the link between moral deontological and consequentialist philosophies and psychological judgments of what is and is not construed as moral.

## Limitations and future directions

The three studies contain several limitations that future research can address. First, we demonstrated and replicated the factor structure of the EPQ-5, but our samples were composed primarily of individuals hailing from English speaking countries. Although past research has repeatedly shown that the original EPQ largely retains its reliability and validity when translated, the EPQ-5 has yet to go through this thorough vetting. We recommend to scholars using the EPQ-5 that along with standard cross-cultural procedures (e.g., back translate; [59]), that the factor structure be examined and, if necessary, demonstrate measurement invariance in new contexts.

Second, we demonstrated convergent and divergent validity with two prominent scales that measure moral foundations and values, but other individual differences such as the five-factor model of personality, dark triad traits, as well as other inventories and scales related to ethics such as the Multidimensional Ethics Scale [60,61] should be examined to determine the EPQ-5's nomological net. In addition, all participants in these studies were administered the EPQ online. Although there is generally little difference between the factor structures, loadings, and the understanding of item content between online participants and those responding in person, differences in mean latent levels have been documented [62].

Third, we cannot speak to the predictive validity of the EPQ-5 regarding ethical behavior. The EPQ has received mixed support for predicting (un)ethical acts and intentions [63–66] and perhaps with the streamlined and robust EPQ-5, greater headway can be made in understanding how ethical position influences ethical intentions and behaviors. Furthermore, we encourage researchers to consider intervening variables (i.e., mediators) to understand this process.

Fourth, although we conducted measurement and structural invariance tests for men and women, there are other key groups of individuals that could be examined. Significant research using the EPQ has tested and found differences in mean levels of idealism and relativism across professions [67], cultures [4], and racial identities [68], and determining whether these differences are attributable to true differences across these groups or are an artifact of the measurement would further establish the robustness or boundary conditions of the EPQ-5.

## Conclusions

We validated and replicated a shortened version of the EPQ labelled the EPQ-5 with two independent samples and demonstrated its convergent and divergent validity. The EPQ-5 was shown to be more psychometrically sound than its original version and is conceptually more aligned with EPT. The taxonomy of EPT was supported and shown to interact with age and sex to predict individual values and moral foundations.

## Supporting information

**S1 Table. The short Ethics Position Questionnaire (EPQ-5).**
(DOCX)

**S2 Table. Study 3 full correlational matrix for variables.**
(DOCX)

**S1 Text. Supplemental text of measurement invariance tests.**
(DOCX)

**S2 Text. Study 3 MANCOVA results for values and moral foundations.**
(DOCX)

## Author Contributions

**Conceptualization:** Donelson R. Forsyth.

**Formal analysis:** Ernest H. O'Boyle, Donelson R. Forsyth.

**Investigation:** Ernest H. O'Boyle, Donelson R. Forsyth.

**Methodology:** Ernest H. O'Boyle, Donelson R. Forsyth.

**Validation:** Ernest H. O'Boyle.

**Writing – original draft:** Ernest H. O'Boyle, Donelson R. Forsyth.

**Writing – review & editing:** Ernest H. O'Boyle, Donelson R. Forsyth.

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
