## [Decision Letter · Decision Letter 0]

6 Apr 2021

PONE-D-21-01338

Individual Differences in Ethics Positions: The EPQ-5

PLOS ONE

Dear Dr. O'Boyle,

Thank you for submitting your manuscript to PLOS ONE. After careful consideration, we feel that it has merit but please check the spelling.

We look forward to receiving your revised manuscript.

Kind regards,

Frantisek Sudzina

Academic Editor

PLOS ONE

Journal Requirements:

'NO. The funders had no role in study design, data collection and analysis, decision to publish, or preparation of the manuscript.'

Reviewers' comments:

Reviewer's Responses to Questions

**Comments to the Author**

1. Is the manuscript technically sound, and do the data support the conclusions?

Reviewer #1: Yes

Reviewer #2: Yes

2. Has the statistical analysis been performed appropriately and rigorously? 

Reviewer #1: Yes

Reviewer #2: I Don't Know

3. Have the authors made all data underlying the findings in their manuscript fully available?

Reviewer #1: Yes

Reviewer #2: Yes

4. Is the manuscript presented in an intelligible fashion and written in standard English?

Reviewer #1: Yes

Reviewer #2: Yes

5. Review Comments to the Author

Reviewer #1: This paper validates and replicates a shorter version of the EPQ-5, addressing issues raised with the original EPQ. I found the paper quite compelling and expect I will make use of this instrument in my own research. I have no substantial criticisms. I did catch a couple of typos: There seems to be something off with the sentence spanning from Page 19 to 20, including that the "of" should presumably be "on", and I think a "be" after "would" and "that" should be "which"? And I think the "and" in the third line from the bottom on Page 22 should be "or".

Reviewer #2: The paper makes an important contribution and the work is clearly and cogently presented, with appropriate contextualization in the scholarly literature. I have just enough familiarity with the core methodologies to be fairly confident of my 'yes' answer to Question 1, on the condition that you have an expert in psychometric analysis address Question 2, to which I answered 'I don't know' because the technicalities are outside the scope of my home discipline.

I enjoyed reading the paper and found the style accessible and pleasingly crisp.

Typographical errors to correct:

p. 7, line 5: 'achieved' should be 'achieve';

p. 7, last line: insert 'of' between 'studies' and 'morality';

p. 20, top line: insert 'be' between 'would' and 'disinclined';

p. 23, 5th line from the bottom, remove extraneous 'The' from 'TheIn' to yield 'In contrast...';

p. 26, I believe 'tenants' should rather be 'tenets';

Other items:

p. 11, spell out abbreviation 'EPT', which occurs twice at p. 11 (or correct to 'EPQ' if that was the intended meaning);

p. 16, insert 'that' between 'predicts' and 'individuals' for easier reading;

p. 26, I would say 'a common practice' rather than 'a widely engaged in practice'

It would be nice to see the name or at least some kind of informative description of the online survey site mentioned at p. 8.

6. PLOS authors have the option to publish the peer review history of their article (what does this mean?). If published, this will include your full peer review and any attached files.

Reviewer #1: No

Reviewer #2: No

---

## [Author Response · Author response to Decision Letter 0]

21 Apr 2021

Response Letter

Thank you for submitting your manuscript to PLOS ONE. After careful consideration, we feel that it has merit but please check the spelling.

Authors’ reply: Thank you very much for the opportunity to revise our manuscript! We have revised it with close attention to the issues raised by you and the reviewer. We feel the manuscript now reads much more smoothly and we’re grateful to you and the review team.

Authors’ reply: Confirmed

We look forward to receiving your revised manuscript.

Authors’ reply: We have now included information regarding financial support from our universities and chairs in the cover letter. 

Kind regards,

Frantisek Sudzina

Academic Editor

PLOS ONE

Journal Requirements:

 Authors’ reply: Confirmed

'NO. The funders had no role in study design, data collection and analysis, decision to publish, or preparation of the manuscript.'

Authors’ reply: Confirmed

a. Please clarify the sources of funding (financial or material support) for your study. List the grants or organizations that supported your study, including funding received from your institution.

d. If you did not receive any funding for this study, please state: “The authors received no specific funding for this work.”

Authors’ reply: We have now added that our universities and endowed chairs supported the project on the title page. We would like to add the amended statement that the funders had no role in study design, data collection and analysis, decision to publish, or preparation of the manuscript.

Authors’ reply: Confirmed

Reviewers' comments:

Reviewer's Responses to Questions

Comments to the Author

5. Review Comments to the Author

Reviewer #1: This paper validates and replicates a shorter version of the EPQ-5, addressing issues raised with the original EPQ. I found the paper quite compelling and expect I will make use of this instrument in my own research. I have no substantial criticisms. I did catch a couple of typos: There seems to be something off with the sentence spanning from Page 19 to 20, including that the "of" should presumably be "on", and I think a "be" after "would" and "that" should be "which"? And I think the "and" in the third line from the bottom on Page 22 should be "or".

Authors’ reply: Thank you very much for your positive feedback on our work. We apologize for the typos and unclear writing in places. We have now carefully read the manuscript to correct the errors you pointed out as well as any others we were able to detect. Again, we greatly appreciate your feedback.

Reviewer #2: The paper makes an important contribution and the work is clearly and cogently presented, with appropriate contextualization in the scholarly literature. I have just enough familiarity with the core methodologies to be fairly confident of my 'yes' answer to Question 1, on the condition that you have an expert in psychometric analysis address Question 2, to which I answered 'I don't know' because the technicalities are outside the scope of my home discipline.

I enjoyed reading the paper and found the style accessible and pleasingly crisp.

Authors’ reply: Thank you very much for your support and guidance!

Typographical errors to correct:

p. 7, line 5: 'achieved' should be 'achieve';

p. 7, last line: insert 'of' between 'studies' and 'morality';

p. 20, top line: insert 'be' between 'would' and 'disinclined';

p. 23, 5th line from the bottom, remove extraneous 'The' from 'TheIn' to yield 'In contrast...';

p. 26, I believe 'tenants' should rather be 'tenets';

Other items:

p. 11, spell out abbreviation 'EPT', which occurs twice at p. 11 (or correct to 'EPQ' if that was the intended meaning);

p. 16, insert 'that' between 'predicts' and 'individuals' for easier reading;

p. 26, I would say 'a common practice' rather than 'a widely engaged in practice'

Authors’ reply: We again apologize for these errors. We have strived to present a fully polished manuscript for maximum readability and free of any mistakes. Thank you again!

It would be nice to see the name or at least some kind of informative description of the online survey site mentioned at p. 8.

Authors’ reply: Thank you for raising this issue. The site included the name of one of the authors and their affiliation. We were trying to preserve the blind review process, but admittedly we could have gone about it in a more direct way (i.e., “website blinded during peer review”). We’ve now adjusted that language and if our paper is accepted we will unmask the URL.

---

## [Decision Letter · Decision Letter 1]

7 May 2021

Individual Differences in Ethics Positions: The EPQ-5

PONE-D-21-01338R1

Dear Dr. O'Boyle,

We’re pleased to inform you that your manuscript has been judged scientifically suitable for publication and will be formally accepted for publication once it meets all outstanding technical requirements.

Kind regards,

Ali B. Mahmoud, Ph.D.

Academic Editor

PLOS ONE

Additional Editor Comments (optional):

Reviewers' comments:

Reviewer's Responses to Questions

**Comments to the Author**

1. If the authors have adequately addressed your comments raised in a previous round of review and you feel that this manuscript is now acceptable for publication, you may indicate that here to bypass the “Comments to the Author” section, enter your conflict of interest statement in the “Confidential to Editor” section, and submit your "Accept" recommendation.

Reviewer #1: All comments have been addressed

Reviewer #2: (No Response)

2. Is the manuscript technically sound, and do the data support the conclusions?

Reviewer #1: Yes

Reviewer #2: Yes

3. Has the statistical analysis been performed appropriately and rigorously? 

Reviewer #1: Yes

Reviewer #2: Yes

4. Have the authors made all data underlying the findings in their manuscript fully available?

Reviewer #1: Yes

Reviewer #2: Yes

5. Is the manuscript presented in an intelligible fashion and written in standard English?

Reviewer #1: Yes

Reviewer #2: Yes

6. Review Comments to the Author

Reviewer #1: I continue to think this is an excellent paper that validates and replicates a shorter version of the EPQ-5, addressing issues raised with the original EPQ. And, again, I find the paper compelling. I recommend publication.

Reviewer #2: Nice work!

A couple of edits still needed at P. 9 in tracked version:

"The EPQ has been used by a several researchers in their studies morality and moral judgment, and the findings are generally support the validity of both the idealism and relativism scale.”

• Please insert ‘of’ between ‘studies’ and ‘morality’

• Please delete ‘are’ between ‘findings’ and ‘generally’

7. PLOS authors have the option to publish the peer review history of their article (what does this mean?). If published, this will include your full peer review and any attached files.

Reviewer #1: No

Reviewer #2: No

---

## [Editor Report · Acceptance letter]

14 May 2021

PONE-D-21-01338R1 

Individual Differences in Ethics Positions: The EPQ-5 

Dear Dr. O’Boyle:

I'm pleased to inform you that your manuscript has been deemed suitable for publication in PLOS ONE. Congratulations! Your manuscript is now with our production department. 

Kind regards, 

on behalf of

Dr. Ali B. Mahmoud 

Academic Editor

PLOS ONE